# Brief communication: Wind speed independent actuator disk control for faster AEP calculations of wind farms using CFD

Maarten Paul van der Laan[1], Søren Juhl Andersen[2], and Pierre-Elouan Réthoré[1]

[1]Technical University of Denmark, DTU Wind Energy, Risø Campus, Frederiksborgvej 399, 4000 Roskilde, Denmark
[2]Technical University of Denmark, DTU Wind Energy, Lyngby Campus, Anker Engelunds Vej 1, 2800 Lyngby, Denmark

**Correspondence:** Maarten Paul van der Laan (plaa@dtu.dk)

**Abstract.** A simple wind speed independent actuator disk control method is proposed that can be applied to speed up annual energy production calculations of wind farms using Reynolds-averaged Navier-Stokes simulations. The new control method allows the user to simulate the effect of different wind speeds in one simulation by scaling a calibrated thrust coefficient curve, while keeping the inflow constant. Since the global flow is not changed, only the local flow around the actuator disks needs be recalculated from a previous converged result, which reduces the number of required iterations and computational effort by a factor of about 2-3.

## 1 Introduction

Wind turbines wakes cause energy losses in wind farms (Barthelmie et al., 2007) and increase blade fatigue loads. The energy losses can be minimized by optimizing the wind farm layout and wind farm control. Fast engineering wake models (Göçmen et al., 2016) are often used to calculate wake effects in optimization routines; however, their accuracy is not guaranteed and higher fidelity wake models are required to validate the annual energy production (AEP) of an optimized wind farm layout. Assessing the AEP for a given wind farm requires in the order of $10^3$ computations to account for all wind speeds and wind directions. Reynolds-averaged Navier-Stokes (RANS) is a relatively fast, yet high-fidelity, computational fluid dynamics method that can be used for this purpose (van der Laan et al., 2015b), although it is still computationally expensive compared to the traditional engineering approaches.

The wind turbines in RANS are often represented by actuator disks (AD), where the wind turbine forces are implemented as a sink term in the momentum equations. The Reynolds number of the RANS-AD simulations, based on a rotor diameter of 100 m, is in the order of $10^7$-$10^8$ for a wind speed range of 4-25 m s$^{-1}$. For these large Reynolds numbers, RANS simulations of a single AD are Reynolds number independent. In other words, the wake deficit, normalized by the inflow wind speed, does not depend on the inflow wind speed, but only depends on the thrust coefficient and atmospheric conditions as ambient turbulence intensity and atmospheric stratification. A proof of the Reynolds number independence of our RANS simulations for a single AD is presented in Appendix A. The Reynolds number independence only holds if the inflow scales by a velocity scale, i.e. the friction velocity. This is true for atmospheric surface layer profiles following Monin-Obukhov Similarity Theory, where the turbulence length scale is also invariant of the wind speed. For RANS-AD simulations of wind farms, the thrust coefficient

varies within the wind farm, which means that one still needs to simulate multiple wind speed cases despite the Reynolds number independence of the flow.

In previous work (van der Laan et al., 2015a), an AD control method was developed for RANS simulations of interacting ADs, where an alternative thrust coefficient $C_T^*$ was used that is dependent on the local AD velocity, averaged over the AD:

$\langle U_{AD} \rangle$. This avoids the necessity of a freestream wind speed (which is not trivial for an AD operating in a wake) to look up the thrust coefficient. The $C_T^*$-$\langle U_{AD} \rangle$ curve was calculated from single RANS-AD simulations. In the present work, the AD control method is made wind speed independent by a simple additional scaling of $\langle U_{AD} \rangle$, as explained in Sect. 2.2. This strategy is related to the work of Andersen (2014), where non-dimensional large-eddy simulations of wind turbines represented by actuator lines were coupled to an aero-elastic model that uses a dimensional controller. The extended AD control method

can be utilized to perform one RANS-AD simulation to simulate the effect of different wind speeds without changing the inflow, thereby reducing the computational effort of a full AEP calculation by a factor of about 2-3 compared to using different inflow wind speeds.

## 2   Methodology

### 2.1   Wind farm simulations

The RANS-AD methodology used for our wind farm simulations are described in previous work (van der Laan et al., 2015b), and a brief summary is presented here. The RANS equation are solved in EllipSys3D, the in-house finite volume flow solver of DTU Wind Energy, initially developed by Sørensen (1994); Michelsen (1992). The $k$-$\varepsilon$-$f_P$ turbulence model is employed (van der Laan et al., 2015c), which has been developed for RANS-AD simulations using a neutral atmospheric surface layer. The 3D flow domain represents a Cartesian domain with dimensions ($L_x = 850D$, $L_y = 830$ and $L_z = 10D$), for the stream-

wise ($x$), lateral ($y$) and vertical directions ($z$), respectively. In the center of flow domain, a uniformly spaced domain is defined with dimensions $54D \times 35D \times 3D$, in which the wind turbines wakes of $5 \times 5$ ADs are resolved. The ADs represent a wind farm of 25 NREL-5MW wind turbines (Jonkman et al., 2009) positioned in a rectangular layout with an inter spacing of 5 rotor diameters $D$. The uniformly spaced domain around the wind farm has a cell size equal to $D/8$, following a grid refinement study of previous work (van der Laan et al., 2015c). The cells sizes are grown while moving away from the wind farm. The

grid consists of 352 blocks of $32^3$ cells, which adds up 11.5 million cells. One CPU per block is used in the simulations (352 CPUs in total). The boundary at $x = 0$ and $z = L_z$ are inlet conditions, at which a logarithmic inflow profile for the streamwise velocity $U$, turbulent kinetic energy $k$ and dissipation of the turbulent kinetic energy $\varepsilon$ is set following Richards and Hoxey (1993). Periodic conditions are imposed on the lateral boundaries and an outflow boundary is set at $x = L_x$, at which all gradients in the streamwise direction are assumed to be zero. The ground is set as a rough wall boundary condition following

Sørensen et al. (2007). An offshore ambient turbulence intensity (based on $k$) of 6 % is set at the wind turbine hub height of 90 m, by using a (uniform) roughness length of $1.9 \times 10^{-4}$ m. The convergence criterion in the RANS-AD simulations is set strict enough to calculate the AEP within a 0.02 % convergence error.

## 2.2 Wind speed independent AD control

The new AD force control method is an extension of previous work (van der Laan et al., 2015a), where the total thrust on the AD is defined by $F_{\text{thrust}} = 1/2 C_T^* \rho A \langle U_{AD} \rangle^2$. Here, $C_T^*$ is the thrust coefficient based on the local AD velocity, averaged over the AD $\langle U_{AD} \rangle$, which can be calculated as $C_T^* = (U_H / \langle U_{AD} \rangle)^2$, with $U_H$ as the freestream wind speed at hub height. The

thrust force distribution of the AD is based on a normalized thrust force distribution computed by a detached eddy simulation of a blade-resolved NREL-5MW rotor for a below-rated wind speed of 8 m s$^{-1}$ (Réthoré et al., 2014). The normalized thrust force distribution is scaled by $C_T^*$ and $\langle U_{AD} \rangle$. Hence, it is assumed that the thrust force distribution, based on a below-rated wind speed, does not change shape. This is generally not true for above-rated wind speeds, where the thrust force distribution is typically more uniform. However, Simisiroglou et al. (2017) has shown that the effect of different thrust force distributions (with

constant total thrust force) on the velocity deficit is mainly visible in the near wake, while the far wake is almost unaffected, especially when atmospheric turbulence is present. In addition, the wake effects above-rated wind speeds are small due to the low thrust coefficient. Hence, the effect of our assumed thrust force distribution on the annual energy production is expected to be small.

Prior to the wind farm simulations, a $C_T^*$-$\langle U_{AD} \rangle$ relation is calculated from a RANS-AD simulation of one AD for each

wind speed between 4 and 25 m/s, for every 1 m/s, using the known $C_T$ curve. We only use one RANS-AD simulation with a constant global inflow, where $C_T$ is updated every time the simulation for a previous $C_T$ has converged. The $C_T^*$-$\langle U_{AD} \rangle$ relation is used in the wind farm simulation to determine the thrust force for each iteration. The power is calculated from a $C_P^*$-$\langle U_{AD} \rangle$ relation, as a post processing step, where $C_P^*$ is defined as $C_P^* = (U_H / \langle U_{AD} \rangle)^3$. If one would like include tangential forces to model wake rotation, it is possible to use a normalized tangential force distribution that is scaled by $C_P^*$ and the tip

speed ratio. In the present work, we do not use tangential forces, because its impact on the power deficit is very small (van der Laan et al., 2015b), and it allows us to use the symmetry of the chosen wind farm layout in order to reduce the number of wind directions necessary to calculate the AEP.

The pre-calculated $C_T^*$-$\langle U_{AD} \rangle$ used in this work is given in Fig. 1, where an additional scaling factor $s$ is added, which is defined as:

$$s = U_H / U_{H,\text{inflow}} \qquad (1)$$

where $U_H$ is the freestream wind speed at hub height that one would like to simulate and $U_{H,\text{inflow}}$ is the actual inflow wind speed at hub height in the RANS-AD wind farm simulation. We multiply $\langle U_{AD} \rangle$ in the $C_T^*$-$\langle U_{AD} \rangle$ by $s$. This simple scaling allows us to perform one RANS-AD wind farm simulation with a fixed inflow profile (e.g. $U_{H,\text{inflow}} = 10$ m s$^{-1}$), and use the scaling parameter $s$ to simulate different wind speeds $U_H$ (e.g. $U_H = 12 \rightarrow s = 1.2$). The control of the start and stop events of

an AD is based on 1D momentum estimate of the freestream velocity, as described in previous work (van der Laan and Abkar, 2019).

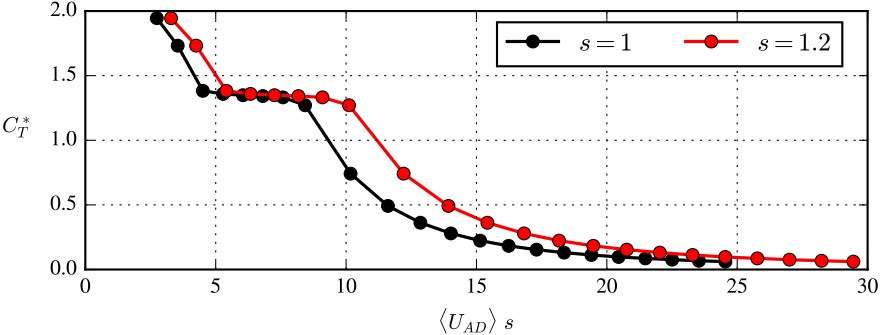

**Figure 1.** Pre-calculated $C_T^*$-$\langle U_{AD} \rangle$ curve used for AD control in wind farm simulations. Scaling parameter $s$ is used to represent different wind speeds.

## 3 Results and Discussion

The AEP of the $5 \times 5$ NREL-5MW wind farm is calculated with RANS-AD simulation(s) using 22 wind speeds between 4 and 25 m s$^{-1}$ (every 1 m s$^{-1}$) and 16 wind directions between 270-315° (every 3°) using the symmetry of the wind farm layout. The wind farm layout is aligned with a wind direction of 270°. A baseline and five additional simulation methodologies are depicted in Fig. 2. The baseline case represents 352 individual simulations, which needs a total of $1.3 \times 10^5$ iterations on the

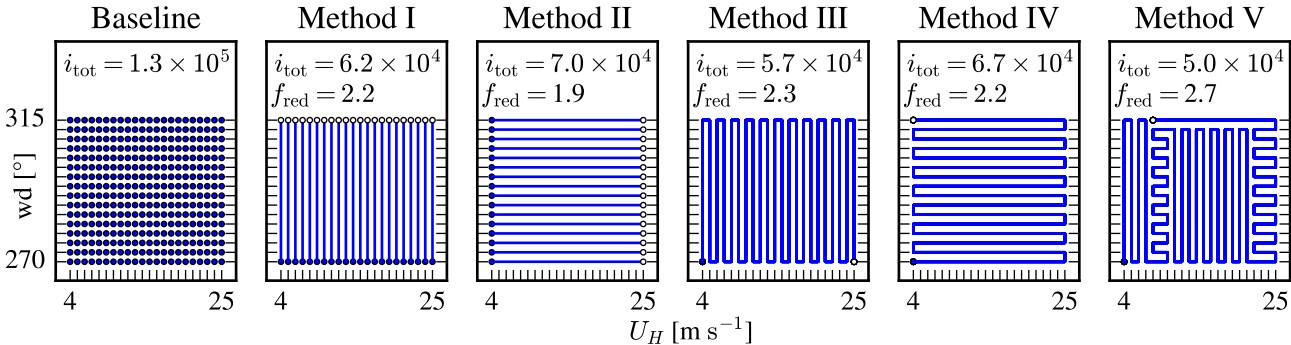

**Figure 2.** Parallel and sequential simulation methodologies to calculate the AEP using 22 wind speeds ($U_H$) and 16 wind directions (wd). Filled and open circles represent the start and end flow cases. $i_{\mathrm{tot}}$ is the number of iteration on the finest grid level and $f_{\mathrm{red}}$ is the reduction of these iterations compared to baseline case.

finest grid ($i_{\mathrm{tot}}$). Each simulation is started with a twice as coarse grid (in all three directions) in order to reduce the number of required iterations on the finest grid. The coarse grid CPU time is negligible compared fine grid CPU time and the total number of iterations on the finest grid is representative for the computational effort. Without the multi-grid approach one

would need 2.4 times more iterations for the baseline case ($3.2 \times 10^5$). The additional methods in Fig. 2 represent sequential or partly sequential simulation(s), where the next wind speed or wind direction is calculated from a previous result. This typically reduces the total number of required iterations because only local flow changes needs to be simulated. However, the required number of iterations is directly proportional to the convergence of the local flow. Therefore, it will not be possible to speed up the computations further by using an initial guess based on *e.g.* one of the fast engineering wake models. The first wind speed and wind direction for Method I-V (filled circles in Fig. 2) is started from a coarser grid level similarly to the Baseline case, while the following wind speed and wind direction are simulated on the finest grid only. The additional methods predict the same AEP within 0.02 % of the Baseline method, which is a result of the chosen convergence criterion. Method I represent 22 separate simulations (one for each wind speed), where the different wind directions are simulated sequentially after each other. The effect of wind direction is simulated by rotating the layout, while keeping the inflow constant. This is possible for RANS-AD simulations using flat terrain and a homogeneous roughness length. The total number of iterations of the finest grid for Method I is reduced by a factor ($f_{\text{red}}$) of 2.2 compared to the Baseline. In Method II, each wind direction is simulated individually and the wind speeds are simulated sequentially using the proposed AD control method from Sect. 2.2, which reduces the number of fine grid iterations by a factor 1.9. Methods III-V are each a single RAND-AD simulation, where all wind speeds and wind directions are simulated by rotation of the layout and scaling of the AD control, respectively. Methods III-V differ in the order of the simulated wind speeds and wind directions. In our presented methods, we choose to either change a wind speed or a wind direction with the smallest step but one could also choose alternative sequential solving paths. This is an optimization problem by itself and it is out of the scope of the present work. Method III and IV provide similar reduction factors, although we have observed that there are differences in the required iterations for particular wind speed and wind direction flow cases. When the thrust coefficient in the wind farm is not varying much (for wind speeds between 7-10 m s$^{-1}$) and well above rated (> 21 m s$^{-1}$), it takes less iterations to perform consecutive wind speeds with respect to consecutive wind directions. The opposite is observed for low wind speeds, when some of the wind turbines experience start up and stop events (i.e. when going from 4 to 5 m s$^{-1}$ or 5 to 4 m s$^{-1}$) and where the thrust coefficient changes rapidly with wind speed (i.e. above the rated wind speed). Method V is a combination of Methods III and IV in order to further reduce the number of fine grid iteration by a factor 2.7 compared to the Baseline case. It should be noted that performance of Method V is dependent on the thrust coefficient curve and wind farm layout.

For RANS-AD simulations including terrain or in-homogeneous roughness lengths, it is not possible to rotate the layout while keeping the inflow direction constant. In this case, our proposed wind speed independent AD control method would reduce the number of required iteration and computational effort significantly following Method II. Note that RANS simulations of non-homogeneous terrain using a logarithmic inflow and rough wall boundary condition are also independent of the Reynolds number, for wind speeds that are relevant for wind turbines, as discussed by Troen et al. (2014). In addition, if one does not have a multi-grid available to reduce the number of fine grid iterations compared to the Baseline case, then the sequential simulation methods (Methods I-V, which do not benefit much from the multi-grid)) yields an additional reduction factor of 2.4.

The total CPU time of the fastest method (Method V) is $1.4 \times 10^{3}$ CPU hours, representing 4 hours in wall clock time. For non-symmetric wind farm layouts, one would need to perform more wind direction cases, i.e. 120 wind directions, which would increase the CPU hours and wall clock time by a factor of about 7.5 representing a wall clock time of about 30 hours using 352 CPUs. By doubling the amount of CPUs one could achieve an AEP calculation of our test wind farm within a day, which makes the RANS-AD model a feasible tool to validate the AEP of a wind farm designed by engineering wake models. It should be noted that if the user has unlimited computational resources available, then the Baseline method is the fastest in terms of wall clock time compared to Methods I-V because all wind speed and wind direction cases can be performed in parallel, which takes only a few minutes per simulation.

In the present work, we have simulated all wind speeds, also far above rated power where the wind turbines do not experience power losses. If one is only interested in the AEP, then these simulations could have skipped once rated wind farm power is achieved. One could further reduce the computational effort of RANS-AD AEP calculations by reducing the number of wind speed and wind direction cases using statistics of the wind resources (i.e. using the Weibull and wind direction distributions), which could be investigated in future work.

Finally, these computations have been performed with an accuracy of 0.02 % for the convergence error in AEP. One could choose to relax the convergence criteria of the solutions at the expense of an higher error, but with a significant reduction in the total amount of computational iterations. For instance, one could choose to accept an error of 0.5 %, which would reduce the computational effort by a factor of 18. An error of 0.5 % would presumably still be less than the uncertainty associated with for instance the wind rose and $C_T$-curve of the turbines. The freed computational resources could then be used to examine these larger uncertainties and as the the numerical error has been quantified, one could correct the final AEP assessment accordingly.

## 4   Conclusions

A simple wind speed independent actuator disk control method is proposed and employed to reduce the number of iterations necessary to calculate the annual energy production from Reynolds-averaged Navier-Stokes simulations of $5 \times 5$ rectangular wind farm with a spacing of 5 rotor diameters. The effect of different wind directions and wind speeds are calculated consecutively in a single simulation by rotating the wind farm layout and using the new wind speed independent actuator disk control method, respectively. Since the global inflow wind speed and wind direction are not changed, only local changes need to be re-calculated for every wind speed and wind direction from a previous converged result, which reduces the total number of iterations by a factor of about 2-3.

*Code and data availability.*   The numerical results are generated with proprietary software, although the data presented can be made available by contacting the corresponding author.

*Author contributions.* MPVDL has performed the simulations, produced all figures and drafted the article. SJA and P-ER have contributed to the methodology and finalization of the paper

*Competing interests.* The authors declare that they have no conflict of interest.

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

## Appendix A: Reynolds number independence of a single wind turbine wake in RANS

The Reynolds number independence of the RANS-AD simulations for a single wind turbine wake is shown in Figure A1, where the stream-wise velocity $U$, normalized by the inflow wind speed at hub height, $U_{H,\text{inflow}}$, is depicted at a downstream distance of five rotor diameters. The same numerical setup is used as discussed in Section 2, where we use a fixed force distribution scaled by $C_T$ and $U_{H,\text{inflow}}$. We have simulated four different cases representing two thrust coefficients ($C_T = 0.4$ and $C_T = 0.8$) and two ambient turbulence intensities at hub height ($I = 5\ \%$ and $I = 10\ \%$) based on the turbulent kinetic energy. Each case is simulated with three different inflow wind speeds, namely, 1, 10 and 100 m s$^{-1}$, corresponding to a Reynolds number of $8.7 \times 10^6$, $8.7 \times 10^7$ and $8.7 \times 10^8$, respectively, for the chosen kinematic molecular viscosity of $1.46 \times 10^{-5}$ m$^2$ s$^1$. Figure A1 clearly shows that normalized velocity deficit is independent of the wind speed.

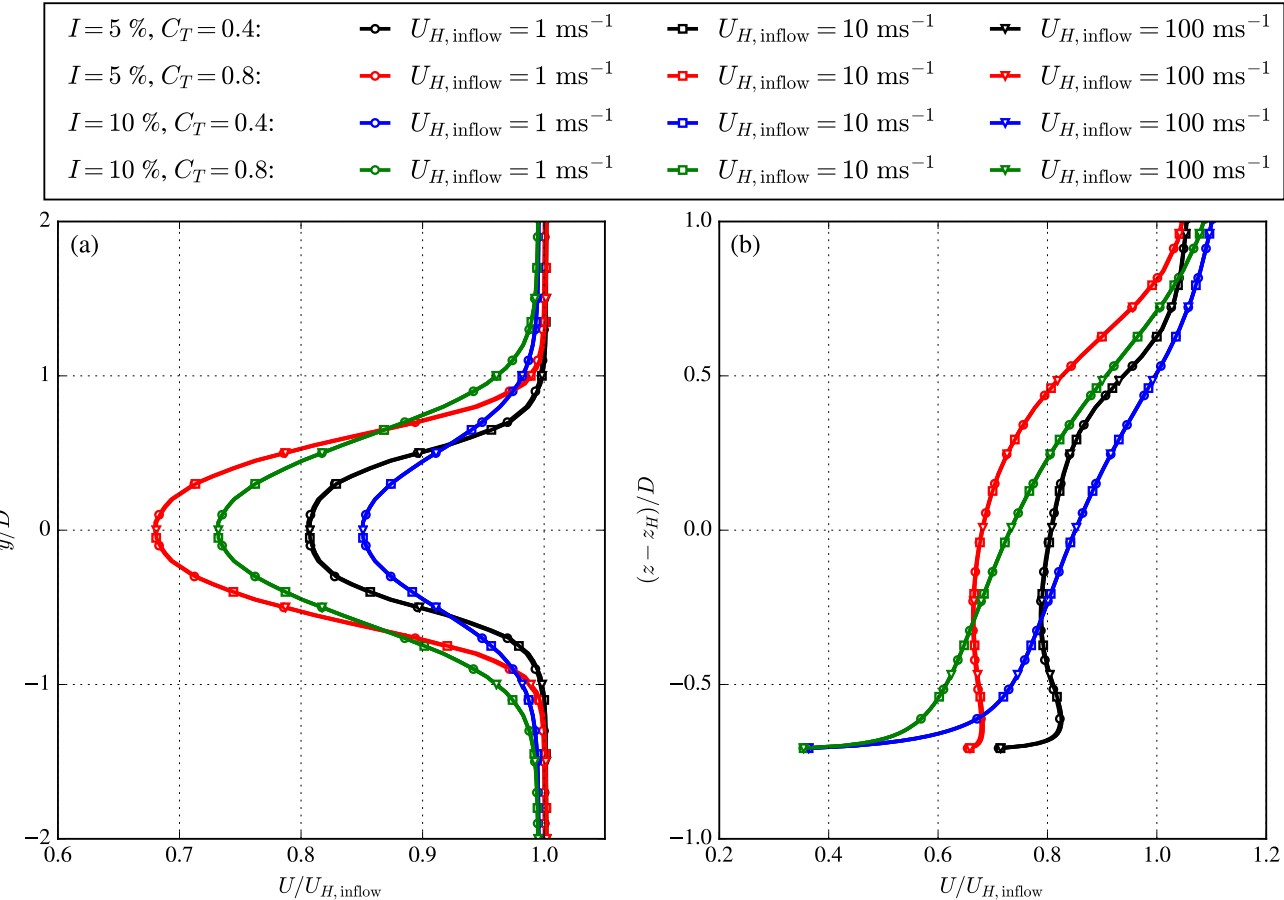

**Figure A1.** Reynolds number independence of the streamwise velocity deficit of a single AD with a prescribed thrust coefficient $C_T$ and ambient turbulence intensity $I$, at downstream distance of five rotor diameters. **(a)** Streamwise velocity at hub height as function of the lateral coordinate. **(b)** Streamwise velocity at $y = 0$ as function of the vertical coordinate.