# Peer review of "Brief communication: Wind speed independent actuator disk control for faster AEP calculations of wind farms using CFD"

_Wind Energy Science, 2019_

## Referee Comment (RC1) · Anonymous Referee #1 · 29 Jun 2019

The manuscript presents a velocity-independent actuator disk method to reduce computational costs for numerical simulations of wind farm flows. The annual energy production (AEP) of a rectangular 5*5 wind farm consisting of NREL-5MW turbines is computed including different wind speeds and directions. The results show that using the new method can reduce computational costs by a factor 2-3.

It is indeed useful to develop RANS simulations with lower computational costs, and I appreciate the authors' efforts to address this issue. However, I have some concerns regarding the methodology suggested in this paper. My main concern is associated with validity of the concept of a velocity-independent actuator disk. Any change in

the incoming velocity can affect wind flow distribution in different ways. In a turbulent boundary layer, any change in the incoming velocity at hub height affects the mean flow shear and size of large turbulent eddies, among other effects. The proposed technique is likely to capture effects of wind speed change on the thrust force. However, it seems that it does not account for changes in flow physics caused by a change in the incoming wind speed. Other more specific comments can be found below:

Page 1, line 20: Apart from the turbulence intensity, the integral length scale is also an important factor as it represents the size of largest turbulent structures in the flow.

Figure 1: Please clarify how the curve shown for a different scaling factor "s" is computed.

Page 3, line 6: How many simulations are performed to estimate Ct*-Uad? Please provide more information about this.

Page 5, line 32: The convergence error is used in this study as the AEP computed by the base case is already available. Please elaborate how this criterion can be used if one aims at using the new technique without a prior knowledge on the true value of AEP.

Minor editorial comments:

Page 1, line 13: I think "relative" should be replaced by "relatively".

Page 1, line 14: "be" is missed in "that can used".

Page 2, line 5: "is" should be removed in "This is strategy ..."

Page 5, line 5: "This an optimization ... " is grammatically incorrect.

---

## Referee Comment (RC2) · Anonymous Referee #2 · 16 Aug 2019

Review of the manuscript:

*M. P. van der Laan, S. J. Andersen and P-E. Réthoré:*
**"Brief communication: Wind speed independent actuator disk control for faster AEP calculations of wind farms using CFD"**

**General comments:**

This brief communication describes the extension of a methodology for calculating the AEP of wind farms. The extension leads to an acceleration of the calculation compared to the baseline method. The description of the basic method along with the introduced simplifications and assumptions can only be comprehended if one is familiar with the corresponding detailed publications referenced by the authors. This is fine for a brief communication. But I suggest that the authors refer to corresponding published works every time when assumptions or simplifications are explicitly mentioned (see also below).

If I understood the manuscript correctly, the novelty of the method concern a scaling of the thrust coefficient $C_T$ to mimic the influence of changes in wind speed instead of actually changing the wind speed in the CFD calculations. This leaves the global flow field unchanged and the modified thrust coefficient results in more local changes of the wind field in the area of the turbines and their wakes. By this and by a clever sequence of restarts based on converged previous calculations, the authors were able to reduce the computation time by a factor of 2-3.

Methods for fast, CFD-based AEP calculation of wind farms are important and work on acceleration of the calculation process are relevant. Therefore, I basically support the publication of this brief communication. However, the description of the method and the new aspects is very difficult for the reader to understand, especially if he does not know the publications of the baseline method. The authors should therefore revise the text taking into account the comments below, describe page 3 in some more detail and include citations at all points where assumptions and simplifications are mentioned.

**Specific comments and remarks:**

- Abstract: The last two sentences of the abstract contain important assumptions of the new aspects of the method and should be picked up at page 3 where the velocity scaling is introduced.

- Introduction: In atmospheric flow properties like integral length scale, turbulence intensity, shear profile etc. depend on wind speed. It is unclear whether these properties are also scaled in the proposed method or if the impact of wind speed is neglected. This should be mentioned and justified.

- 2.2, l.2-3 p.3: Unless the reader already knows the cited previous work, it is unclear that the average of the square velocity is used to obtain the scaled thrust coefficient $c_T^*$. Please add shortly this information.

- 2.2: l. 4-5 p.3: "The thrust force distribution of the AD is based on a normalized thrust force distribution". For the NREL 5 MW wind turbine, the thrust force distribution almost linearly scales with the rotor thrust coefficient $c_T$ only below rated conditions and is flattened at higher wind speeds. Please add some information about how the thrust distribution is scaled with your method and how you deal with above rated situations.

- 2.2: l. 6 p.3: Please define what is meant with "standard $C_T$ curve" and give some reference.

- 2.2, p.3: It is unclear to me how the scaling parameter s is used within the simulation. Please clarify and give some justification.

- Conclusions: l.11-12 p.6: The application of this method to complex terrain situations should first be proven. In complex terrains, flow inclination, changes of the wind direction over the rotor disc, flow separation and large scale turbulent structures are apparent. These effects do not necessarily linearly scale with the inflow velocity. I am looking forward to your results.

---

## Author Comment (AC1) · 12 Sep 2019

The comment was uploaded in the form of a supplement:
https://www.wind-energ-sci-discuss.net/wes-2019-27/wes-2019-27-AC1-supplement.pdf

---

## Author Response (AR1)

**Reply to reviewers**

September 12, 2019

We would like to thank the two reviewers for their feedback and suggestions to improve the article. In the proceeding sections, the reviewers comments are copied and answered per comment (blue color). An additional document is provided that highlights all modifications with respect to the initial submitted version.

**Reviewer 1**

The manuscript presents a velocity-independent actuator disk method to reduce computational costs for numerical simulations of wind farm flows. The annual energy production (AEP) of a rectangular 5*5 wind farm consisting of NREL-5MW turbines is computed including different wind speeds and directions. The results show that using the new method can reduce computational costs by a factor 2-3. It is indeed useful to develop RANS simulations with lower computational costs, and I appreciate the authors' efforts to address this issue. However, I have some concerns regarding the methodology suggested in this paper. My main concern is associated with validity of the concept of a velocity-independent actuator disk. Any change in the incoming velocity can affect wind flow distribution in different ways. In a turbulent boundary layer, any change in the incoming velocity at hub height affects the mean flow shear and size of large turbulent eddies, among other effects. The proposed technique is likely to capture effects of wind speed change on the thrust force. However, it seems that it does not account for changes in flow physics caused by a change in the incoming wind speed.

We do not share this concern because the inflow turbulence length scale $\ell$ in our RANS setup follows the neutral atmospheric surface layer: $\ell = \kappa z$, with $\kappa$ as the Von Karman constant and $z$ as the height. This turbulence length scale is independent of the wind speed. The turbulence length scale in wind turbine wake deviates from the inflow turbulence length scale, but the velocity deficit normalized by a inflow wind speed, is not dependent on the inflow wind speed. We understand from the review process that this concept might not be well know. Therefore, we have now added an Appendix showing a proof of the Reynolds number independence of the RANS simulations of a single wind turbine wake in a neutral atmospheric surface layer, for a wind speed at hub height of 1, 10 and 100 m/s, for two thrust coefficients and two turbulence intensities. These wind speeds correspond to the following Reynolds numbers based on the rotor diameter: $8.7 \times 10^6$, $8.7 \times 10^7$ and $8.7 \times 10^8$ (where the kinematic molecular viscosity is set to $1.46 \times 10^{-5}$ m² s¹).

Other more specific comments can be found below:

1. Page 1, line 20: Apart from the turbulence intensity, the integral length scale is also an important factor as it represents the size of largest turbulent structures in the flow.
   See previous answer.

2. Figure 1: Please clarify how the curve shown for a different scaling factor "s" is computed.
   We use the scaling factor $s$ from equation 1, to multiply $\langle U_{AD} \rangle$ in the $Ct$-$\langle U_{AD} \rangle$ relation. Figure 1 shows this for $s = 1.2$. We have added the following in Section 2.2: "We multiply $\langle U_{AD} \rangle$ in the $C_T^*$-$\langle U_{AD} \rangle$ by $s$."

3. Page 3, line 6: How many simulations are performed to estimate Ct*-Uad? Please provide more information about this.
   We have added: "Prior to the wind farm simulations, a $C_T^*$-$\langle U_{AD} \rangle$ relation is calculated from a RANS-AD simulation of one AD for each wind speed between 4 and 25 m/s, for every 1 m/s, using the known

$C_T$ curve. We only use one RANS-AD simulation with a constant global inflow, where $C_T$ is updated every time the simulation for a previous $C_T$ has converged."

4. Page 5, line 32: The convergence error is used in this study as the AEP computed by the base case is already available. Please elaborate how this criterion can be used if one aims at using the new technique without a prior knowledge on the true value of AEP.
   The convergence error in the AEP was calculated using several AEP calculations with different levels of convergence criteria. In this work, we have only looked at one wind farm layout, one wind turbine type and one level of turbulence intensity. The change of one of these may lead to the need for stricter convergence criterium. We normally set a quite conservative level of convergence. One could investigate how changes in wind farm layout, wind turbine type and turbulence intensity affect the convergence criterium, but this is out of the scope of the present work. We have discussed the level of convergence in Section 5 because it has a high impact on the computational cost. If one is interested in a rough estimate of the AEP, a less strict convergence criterium could be considered to get a quick result.

5. Minor editorial comments:

   (a) Page 1, line 13: I think "relative" should be replaced by "relatively".
   (b) Page 1, line 14: "be" is missed in "that can used".
   (c) Page 2, line 5: "is" should be removed in "This is strategy ..."
   (d) Page 5, line 5: "This an optimization ... " is grammatically incorrect.

   We have corrected these four typos.

**Reviewer 2**

This brief communication describes the extension of a methodology for calculating the AEP of wind farms. The extension leads to an acceleration of the calculation compared to the baseline method. The description of the basic method along with the introduced simplifications and assumptions can only be comprehended if one is familiar with the corresponding detailed publications referenced by the authors. This is fine for a brief communication. But I suggest that the authors refer to corresponding published works every time when assumptions or simplifications are explicitly mentioned (see also below).

If I understood the manuscript correctly, the novelty of the method concern a scaling of the thrust coefficient CT to mimic the influence of changes in wind speed instead of actually changing the wind speed in the CFD calculations. This leaves the global flow field unchanged and the modified thrust coefficient results in more local changes of the wind field in the area of the turbines and their wakes. By this and by a clever sequence of restarts based on converged previous calculations, the authors were able to reduce the computation time by a factor of 2-3.

Methods for fast, CFD-based AEP calculation of wind farms are important and work on acceleration of the calculation process are relevant. Therefore, I basically support the publication of this brief communication. However, the description of the method and the new aspects is very difficult for the reader to understand, especially if he does not know the publications of the baseline method. The authors should therefore revise the text taking into account the comments below, describe page 3 in some more detail and include citations at all points where assumptions and simplifications are mentioned.

Specific comments and remarks:

1. Abstract: The last two sentences of the abstract contain important assumptions of the new aspects of the method and should be picked up at page 3 where the velocity scaling is introduced.
   We have extended the description of the new AD control method in Section 2.2.

2. Introduction: In atmospheric flow properties like integral length scale, turbulence intensity, shear profile etc. depend on wind speed. It is unclear whether these properties are also scaled in the proposed method or if the impact of wind speed is neglected. This should be mentioned and justified.
   Reviewer 1 had a very similar comment about this. Please find the answer in the reply to Reviewer 1.

3. 2.2, l.2-3 p.3: Unless the reader already knows the cited previous work, it is unclear that the average of the square velocity is used to obtain the scaled thrust coefficient cT*. Please add shortly this information.
We added how $C_T^*$ is calculated: "..., which can be calculated as $C_T^* = (U_H/\langle U_{AD} \rangle)^2$, with $U_H$ as the freestream wind speed at hub height." In addition, we have also added how $C_P^*$ is calculated.

4. 2.2: l. 4-5 p.3: "The thrust force distribution of the AD is based on a normalized thrust force distribution". For the NREL 5 MW wind turbine, the thrust force distribution almost linearly scales with the rotor thrust coefficient cT only below rated conditions and is flattened at higher wind speeds. Please add some information about how the thrust distribution is scaled with your method and how you deal with above rated situations.
The AD method from van der Laan et al. (2015), assumes that the normalized force distribution does not change with wind speed, and this assumption is also made in the present work. The reviewer is right that the thrust force distribution is different above rated wind speeds and our assumption is violated. One could argue that the wake effects above rated are rather small due to the low thrust coefficient. Hence, the impact of our constant normalized thrust force distribution is expected to be small. In addition, Simisiroglou et al. (2017) has shown that the effect of different force distributions (for the same total thrust force) has only an impact on the near wake, while the far wake is very similar, especially when atmospheric turbulence is included. We have added this discussion to Section 2.2.

5. 2.2: l. 6 p.3: Please define what is meant with "standard CT curve" and give some reference.
We have changed "standard" to "known" $C_T$ curve to make this more clear.

6. 2.2, p.3: It is unclear to me how the scaling parameter s is used within the simulation. Please clarify and give some justification.
The scaling parameter $s$ is used to multiply $\langle U_{AD} \rangle$ in the $Ct$-$\langle U_{AD} \rangle$ relation. Figure 1 shows this for $s = 1.2$. We have the following in Section 2.2: "We multiply $\langle U_{AD} \rangle$ in the $C_T^*$-$\langle U_{AD} \rangle$ by $s$."

7. Conclusions: l.11-12 p.6: The application of this method to complex terrain situations should first be proven. In complex terrains, flow inclination, changes of the wind direction over the rotor disc, flow separation and large scale turbulent structures are apparent. These effects do not necessarily linearly scale with the inflow velocity. I am looking forward to your results.
For RANS simulations of complex terrain using a logarithmic inflow, the speed up factor is independent of the wind speed due to the Reynolds number independence. Note that the viscous sub layer near the wall is not resolved since we use a rough wall boundary condition. We have added a reference discussing the Reynolds number independence: Troen et al. (2014) in Section 3. The reviewer could also have a look at a presentation given by Bechmann (2014), where the Reynolds number independence was shown by RANS simulations of the speed up factor over complex terrain.
The reviewer is right that we have not applied the new wind speed independent AD control method in the present work. Therefore, we have removed the statement about the application to complex terrain in the conclusion.

[revised manuscript text omitted]

---

## Referee Report (RR1)

Review of the revised manuscript:

*M. P. van der Laan, S. J. Andersen and P-E. Réthoré:*
**"Brief communication: Wind speed independent actuator disk control for faster AEP calculations of wind farms using CFD"**

The authors are thanked for considering my remarks and for the revision of the manuscript. The paper is now much more comprehensible. The clarification that the Reynolds number independence applies only to the specific turbulence model used and the related verification in the appendix is important to understand and accept the suggested approach. I support publication of the submission without further changes. Please note the small typo at p. 3, line 7: "Hence, it assumed" → "Hence, it is assumed…"